# Hydrodynamic Analysis of a Modular Floating Structure with Tension-Leg Platforms and Wave Energy Converters

**Nianxin Ren [1,2], Hongbo Wu [3], Kun Liu [4,\*], Daocheng Zhou [3] and Jinping Ou [3]**

[1]  College of Civil Engineering and Architecture, Hainan University, Haikou 570228, China; rennianxin@hainanu.edu.cn

[2]  State Key Laboratory of Marine Resource Utilization in South China Sea, Hainan University, Haikou 570228, China

[3]  State Key Laboratory of Coast and Offshore Engineering, Dalian University of Technology, Dalian 116024, China; 18929523984@163.com (H.W.); zhoudc@dlut.edu.cn (D.Z.); oujinping@dlut.edu.cn (J.O.)

[4]  School of Civil Engineering and Transportation, South China University of Technology, Guangzhou 510641, China

\*  Correspondence: liukun86@scut.edu.cn

**Abstract:** This work presents a modular floating structure, which consists of five inner tension-leg platforms and two outermost wave energy converters (denoted as MTLPW). The hydrodynamic interaction effect and the mechanical coupling effect between the five inner tension-leg platforms (TLP) and the two outermost wave energy converters (WEC) are taken into consideration. The effects of the connection modes and power take-off (PTO) parameters of the WECs on the hydrodynamic performance of the MTLPW system are investigated under both operational and extreme sea conditions. The results indicate that the hydrodynamic responses of the MTLPW system are sensitive to the connection type of the outermost WECs. The extreme responses of the bending moment of connectors depend on the number of continuously fixed modules. By properly utilizing hinge-type connectors to optimize the connection mode for the MTLPW system, the effect of more inner TLP modules on the hydrodynamic responses of the MTLPW system can be limited to be acceptable. Therefore, the MTLPW system can be potentially expanded to a large degree.

**Keywords:** modular system; TLP; WEC; hydrodynamic interaction; VLFS

## 1. Introduction

As coastal land resources become more and more precious due to the increasing urban population, the very large floating structure (VLFS) has been paid more and more attention for its environmentally friendly and cost-effective characteristics. A VLFS can be applied to create "land" on the sea for various applications, such as a floating airport [1,2], floating oil storage facility [3], and floating fish farm [4]. However, researchers have pointed out that VLFS has to meet the challenge of significant structural bending moment and deformation due to its huge scale. Therefore, the hydro-elastic analysis should be well concerned for the safety design of a VLFS [5].

Various approaches for the hydro-elastic analysis of the VLFS have been proposed. The most common method is to treat the VLFS as a floating beam or a thin plate, simplifying the hydro-elastic analysis with linear potential theory [6–9]. To further improve the resolution of the numerical model, the three-dimensional linear hydro-elastic model has been proposed [10,11]. One of the most effective methods is to divide the VLFS into several modules which are connected by rigid or flexible connectors [12–14], which can make it much easier to construct, transport, install, and extend the VLFS. Compared with rigid connectors, semirigid or flexible hinge connectors among the adjacent modules can be helpful for mitigating the hydro-elastic responses of the VLFS [15,16]. The stiffness of connectors has great influence on the hydro-elastic response of the VLFS, especially for the

connection loads [17,18]. The modular VLFS system can be regarded as a rigid-module-flexible-connector (RMFC) model, and a RMFC model of four connected semisubmersible platforms by mechanical joints has been proposed [19]. In addition, a comparative study showed that the RMFC model is less time-consuming than the finite element analysis (FEA) model, and it can also predict the hydrodynamic responses of the mobile offshore base (MOB) very well [20]. Therefore, the RMFC model seems more suitable for the preliminary and conceptual design of the modular VLFS.

A nonlinear dynamic network model for the modular VLFS has been proposed, which can account for different nonlinear effects such as sudden changes of module responses [21], amplitude death [22–24], and collective behaviors [25]. The availability of this nonlinear numerical model has been verified by corresponding model tests [26]. The results indicate that the outermost modules of the modular VLFS usually suffer larger motion amplitudes than the inner ones, especially for the outermost flexible connectors [27]. If the outermost flexible connectors can be equipped with WEC power take-off (PTO) dampers, the motion responses of the outermost modules can not only be effectively reduced but considerable wave energy power can also be obtained [28,29]. In addition, some researchers [30,31] have tried to combine floating offshore wind turbines with Oscillating Water Column (OWC) devices, and some promising findings of the new hybrid wind-wave energy systems have been pointed out.

So far, most modular VLFSs are pontoon-type or semisubmersible, but the uneven vertical carrying loads on adjacent modules may have a strong impact on the draft difference among modules, which tends to significantly increase connectors' vertical loads. Consequently, the variable vertical carrying capacity of adjacent modules would be limited to protect their connectors. However, if the VLFS can be modularized by TLPs, there will be almost no draft change for different vertical carrying loads on adjacent modules, due to the large pretension force of tension legs. Ren [32] proposed a novel modular floating structure system with tension-leg platforms, which can effectively relieve the connectors' vertical loads and enhance the modules' variable carrying capacity.

In the present work, a modular floating structure with five inner TLPs and two outermost WECs (denoted as MTLPW) is proposed. The hydrodynamic responses of the MTLPW system are investigated under typical sea conditions. Both the hydrodynamic interaction effect and the mechanical coupling effect among seven modules are well considered. The effects of different connection modes and WECs' PTO parameters on the hydrodynamic performance of the proposed system are investigated.

## 2. Numerical Model of the MTLPW

### 2.1. Description of the MTLPW

A modular floating structure system with inner TLPs and outermost WECs (denoted as MTLPW [32]) was proposed for potential applications of the cost-effective 'land on sea', which is shown in Figure 1. The outermost WEC modules can both reduce the wave loads acting on inner TLP modules and utilize relative pitch motions to produce wave energy. The dimensions of inner TLP modules and WEC modules are the same, and the number of these involved standardized modules of the MTLPW system can be flexibly adjusted to the desired full-scale dimensions and functions. Ships can access the side without WEC modules.

It should be noticed that the paper mainly focuses on the preliminary feasibility study of the MTLPW system. To balance the accuracy of the numerical model and corresponding computational time, a simplified seven-module connected MTLPW system is proposed for the investigation of its hydrodynamic characteristics in a mild sea zone (usually with the protection of natural islands or artificial seawalls [32]).

The sketch of the simplified 7-module MTLPW system (in side view) is shown in Figure 2. The proposed system includes 5 inner TLP modules and 2 outermost WEC modules:

(1)   Each TLP module was initially designed to withstand about 2000t mass variation (available for TLP pretension range of 3000~5000 t), with 4 tension legs symmetrically

distributed at the 4 corners and several anticollision devices installed at the bottom-corners. The anticollision devices serve as fenders with linear springs to monitor possible bottom impact forces.

(2) Two outermost pitch-type WEC modules (M1 and M7) symmetrically connect to their adjacent outermost TLP modules (M2 and M6), respectively. The 2 outermost hinge-type connectors (denoted as C1 and C6) are coupled with WEC PTO systems, which can serve as linear hydraulic dampers to utilize the relative pitch motion between the outermost WEC module and its adjacent TLP module to produce power. It should be pointed out that different PTO damping levels (damping coefficients) of the WEC can be flexibly adjusted using the throttle valve in practice.

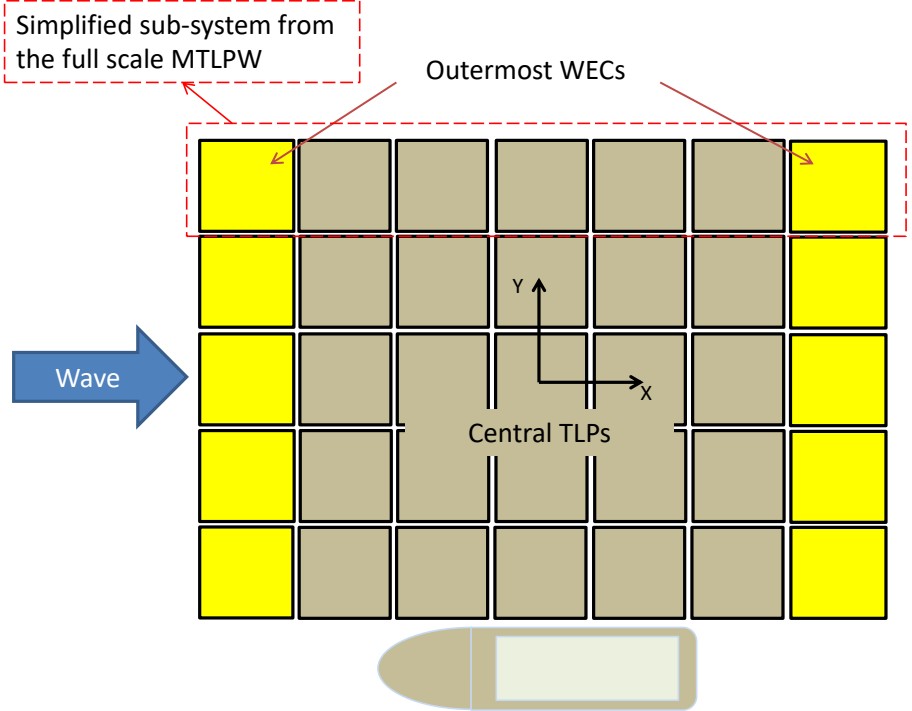

**Figure 1.** Sketch of the conceptual modular floating structure, which consists of 5 inner tension-leg platforms and 2 outermost wave energy converters (denoted as MTLPW) (in top view).

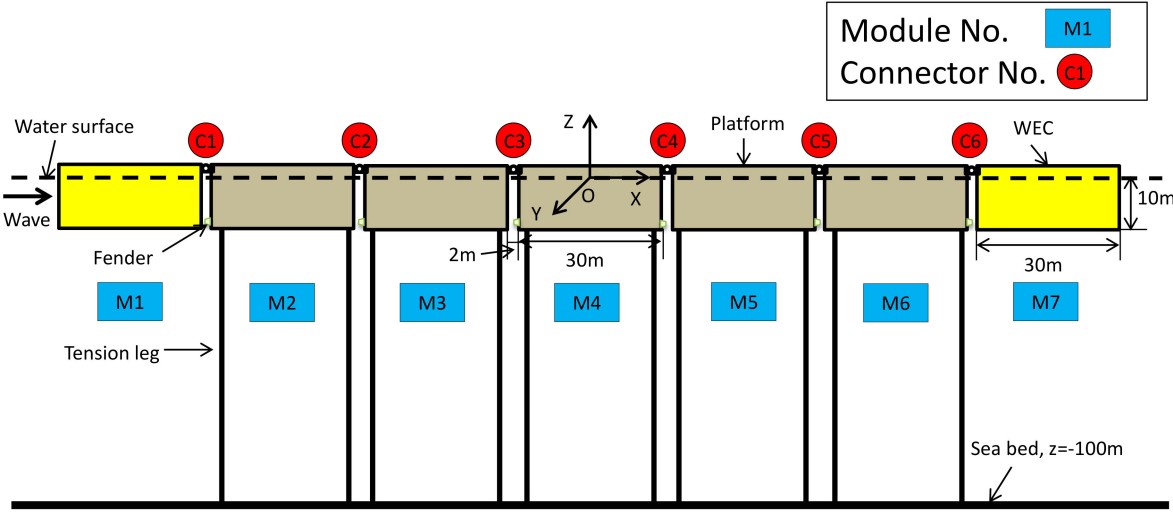

**Figure 2.** Sketch of the simplified 7-module MTLPW system (in side view).

The effects of both the WEC modules and the connection modes on the hydrodynamic responses of the simplified 7-module MTLPW system in a mild sea zone and the key design parameters are listed in Table 1.

**Table 1.** Main design parameters of the simplified 7-module MTLPW [32].

| Parameters | Value | Units |
|---|---|---|
| **Inner TLP module** | | |
| Dimension | $30 \times 30 \times 12$ | m |
| Draft | 10.0 | m |
| Water depth | 100.0 | m |
| Mass; Displacement | 4225; 9225 | t |
| Total TLP pretension | 5000 | t |
| Height of mass center | $-7.34$ | m |
| Ixx = Iyy, Izz | $6.0 \times 10^8$; $6.8 \times 10^8$ | Kg·m$^2$ |
| Tension leg dimension | D = 1.2; t = 0.04; L = 90 | m |
| Steel tension leg E; $\sigma_s$ | $2.1 \times 10^{11}$; $3.45 \times 10^8$ | N/m$^2$ |
| Stiffness of bottom fenders | $1.0 \times 10^7$ | N/m |
| **Outer WEC module** | | |
| Dimension | $30 \times 30 \times 12$ | m |
| Draft | 10.0 | m |
| Mass = displacement | 9225 | t |
| Height of mass center | $-4$ | m |
| Ixx = Iyy, Izz | $1.4 \times 10^9$, $2.0 \times 10^9$ | kg·m$^2$ |
| Adjacent distance | 2.0 | m |
| WEC PTO damping | $5.0 \times 10^8$ | Nms/rad |

### 2.2. Governing Equation

In this paper, the simplified MTLPW system is viewed as a RMFC model, which means that each module of the combined system is modeled as a rigid body, and the structural deformation mainly happens in the connectors among modules. Thus, the government equation of the simplified 7-module MTLPW system can be generally summarized as follows:

$$M_i \ddot{X}_i + C_i \dot{X}_i + K_i(X_i) = F_{i,Wave} + F_{i,Con} + F_{i,tlp} + F_{i,fender} \tag{1}$$

where $X_i$ (6 Degree of Freedom, 6-DOF) indicates the generalized displacement vector of the *i*-th module. $M_i$, $C_i$, and $K_i$ denote the mass matrix, the damping matrix, and the hydrostatic restoring matrix, respectively. $C_i$ is the commonly artificial damping to compensate the viscous fluid effect. $F_{i,\text{wave}}$, $F_{i,\text{con}}$, $F_{i,\text{tlp}}$, and $F_{i,\text{fender}}$ are the generalized wave force matrix, the connector force matrix, the tension matrix for tension legs, and the possible bottom fender impact force matrix, respectively. The subscript number *i* (*i* = 1~7) of each matrix indicates the *i*-th standardized module along the incident wave direction.

### 2.3. Hydrodynamic Model

The hydrodynamic model of the simplified 7-module MTLPW system is established based on the AQWA code [33], which is available and widely used for the hydrodynamic analysis of the TLP, the multibody hydrodynamic interaction effect, and the mechanical coupling effect [32]. The hydrodynamic numerical model of the simplified 7-module MTLPW system is shown in Figure 3.

Seven modules are involved in the hydrodynamic model, so the hydrodynamic interaction effect among modules has to be considered. The total velocity potential can be generally written as follows:

$$\phi = \phi_I + \phi_D + i\omega \sum_{i=1}^{7} \sum_{j=1}^{6} u_i^j \phi_i^j \tag{2}$$

where $\phi_I$ and $\phi_D$ indicate potential of incident and diffraction, respectively. $u_i^j$ is the complex amplitude of the $i$-th module in the $j$-th modal (6-DOF). $\phi_i^j$ is the potential that is only caused by a unit amplitude motion of the $i$-th module, which views other modules as fixed, indicating the normalized velocity potential of the $j$-th modal of the $i$-th module.

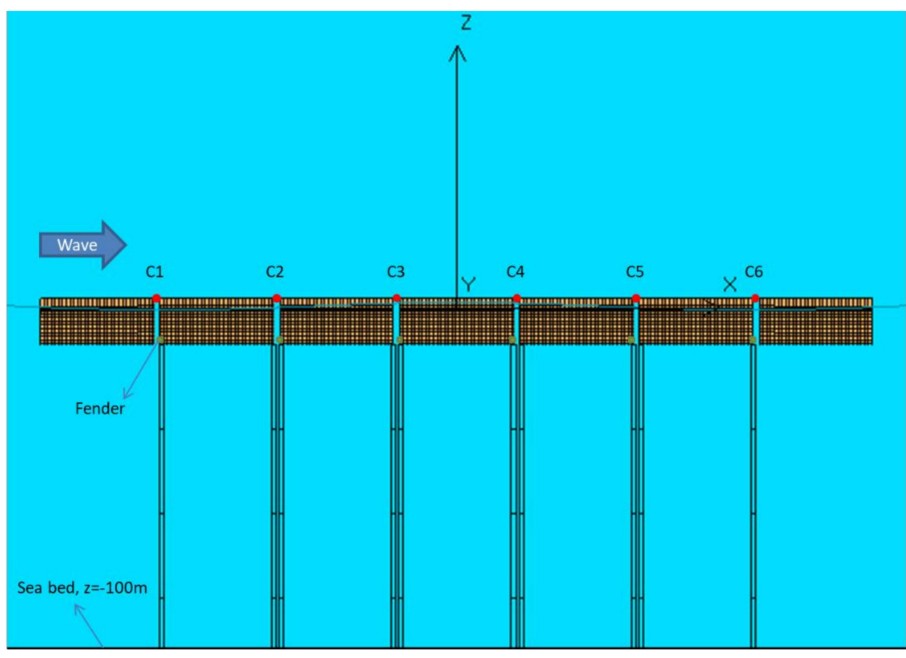

**Figure 3.** Hydrodynamic model of the simplified 7-module MTLPW system.

Then, by integrating wave velocity potential along the wet surface of the $i$-th module, the wave force can be expressed as follows:

$$
\begin{aligned}
F_{i,Wave} &= \iint_{S_i} n_i [-i\omega\rho(\phi_I + \phi_D + i\omega \sum_{i=1}^{7} \sum_{j=1}^{6} u_i^j \phi_i^j)] ds \\
&= -i\omega\rho \iint_{S_i} n_i (\phi_I + \phi_D) ds - \sum_{i=1}^{7} \sum_{j=1}^{6} (\omega^2 \rho \iint_{S_i} \mathrm{Re}(\phi_i^j) n_i ds - i\omega^2 \rho \iint_{S_i} \mathrm{Im}(\phi_i^j) n_i ds u_i^j) \\
&= F_{i,w} - \sum_{i=1}^{7} \sum_{j=1}^{6} (A_{ij}^j \ddot{u}_i^j + B_{ij}^j \dot{u}_i^j) u_i^j
\end{aligned}
\tag{3}
$$

where the wave forces are split into 2 parts. $F_{i,w}$ indicates the wave exciting force because of the scattering potential ($\phi_I$ and $\phi_D$), and the other part indicates the radiation force caused by the interaction of seven modules' radiation potential. $A_{ij}$ and $B_{ij}$ indicate the terms of added mass and added damping, respectively.

The 7 modules are assumed to stay in harmonic motion excited by wave forces, so the total wave force on the $i$-th module can be calculated as follows:

$$
F_{i,Wave} = F_{i,w} e^{-i\omega t} - \sum_{j=1}^{7} (A_{ij} \ddot{U}_i + B_{ij} \dot{U}_i)
\tag{4}
$$

In addition, the total tension leg force on the $i$-th module can be calculated as follows:

$$
F_{i,tlp} = \sum_{j=1}^{4} E_i A_i \varepsilon_{ij}
\tag{5}
$$

where $\varepsilon_{ij}$ is the strain of the $j$-th tension leg of the $i$-th module. $E_i$ and $A_i$ are the elasticity modulus and the section area of the tension leg of the $i$-th module.

*2.4. Connector Types*

The MTLPW system involves 3 types of connectors, which are listed as follows:

(1) The fixed connector (denoted as Fixed): There is no relative motion in all degrees of freedom between 2 adjacent connected modules.

(2) The hinge connector (denoted as Hinge): There is only freely relative pitch motion between 2 adjacent connected modules.

(3) The hinge connector with an additional WEC linear pitch damper (Kp) (denoted as HWK): The additional damper can mitigate the relative pitch motion of 2 adjacent modules to some degree, and serves as the PTO system of the WEC module to generate power. The original pitch damping was designed to be $5.0 \times 10^8$ Nms/rad, and the design parameters for each HWK connector in the MTLPW system are the same. In addition, the HWK connector type for wave power production was only considered for the 2 outermost connectors (C1 and C6).

The connector forces ($F_{i,con}$) acting on the *i*-th module induced by adjacent modules can be expressed as:

$$F_{i,Con} = \sum_{j=1}^{7} \left( \varphi_{ij} Kc_{ij} \delta(X_i, X_j) \right) \tag{6}$$

where $\varphi_{ij}$ is a topology matrix. $\varphi_{ij}$ is set to be 1 when the *j*-th module connects with the *i*-th module. Otherwise, $\varphi_{ij}$ is set to be 0. $Kc_{ij}$ is the connection stiffness matrix between the *i*-th module and the *j*-th module. $\delta(X_i, X_j)$ is the relative motion matrix between the *i*-th module and the *j*-th module.

In addition, the possible bottom fender impact force $F_{i,fender}$ can be simplified estimated as follows:

$$F_{i,fender} = \begin{cases} Kf_{ij} \cdot \delta x(X_i, X_j) & \text{if } \delta x(X_i, X_j) < -2 \ m \text{ (contact)} \\ 0 & \text{if } \delta x(X_i, X_j) \geq -2 \ m \text{ (no contact)} \end{cases} \tag{7}$$

where $Kf_{ij}$ ($1.0 \times 10^7$ N/m) is the bottom fender linear stiffness coefficient between the *i*-th module and the adjacent *j*-th module. $\delta x(X_i, X_j)$ is the relative bottom surge motion between the i-th module and the adjacent *j*-th module. If the negative relative bottom surge motion $\delta x(X_i, X_j)$ is smaller than the module's gap (2 m), the 2 adjacent modules will be in bottom contact and the bottom fender contact force will be observed.

*2.5. Estimation of Wave Power Production*

For the 2 outermost HWK connectors (C1 and C6), the WEC PTO systems were simplified as linear pitch dampers ($Kp$). The bending moment of the pitch damper ($M_{Bpto}$) and the corresponding relative pitch velocity ($w_{ref}$) between the WEC module and the adjacent TLP module can be used to estimate the power production of the WEC ($P_{wave}$). The corresponding formula can be written as follows:

$$P_{wave}(t) = M_{Bpto}(t) \cdot w_{ref} = M_{Bpto}^2(t) / Kp \tag{8}$$

## 3. Numerical Results

This work mainly focuses on both the effects of different connection modes of the simplified seven-module MTLPW system and the WEC module on the hydrodynamic performance of the proposed system.

*3.1. Effects of Different Connection Modes*

Considering that the pitch response of the WEC module has significant influence on both the energy production and the dynamic responses of the proposed system, the pitch RAO (Response Amplitude Operator) of a free WEC module was investigated, as shown in Figure 4. It can be seen that the natural pitch period of the free WEC module was around 10 s. In order to optimize the performance of the WEC modules, one typical regular sea

case (*H* = 2 m, *T* = 10 s) was selected for the initial investigation of the hydrodynamic responses of the seven-module MTLPW system with different connection modes. Five connection modes for the seven-module MTLPW system were considered, which are listed in Table 2. The symbols "H," "f," and "h" denote the HWK connector, fixed connector, and hinge connector, respectively. The symbol "-" was used to link adjacent connectors in the seven-module MTLPW system. The surge response is usually the most significant motion of the TLP module [34–36], while the heave and pitch responses are both important for the outermost WEC modules. Therefore, the corresponding motion responses of the seven-module MTLPW system are shown in Figure 5, and the amplitudes of the connectors' loads are shown in Figure 6.

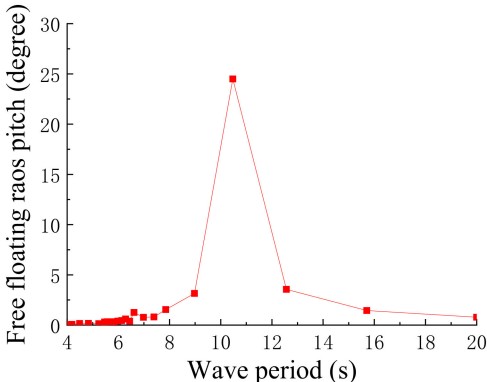

**Figure 4.** Pitch RAO of the free wave energy converters (WEC) module.

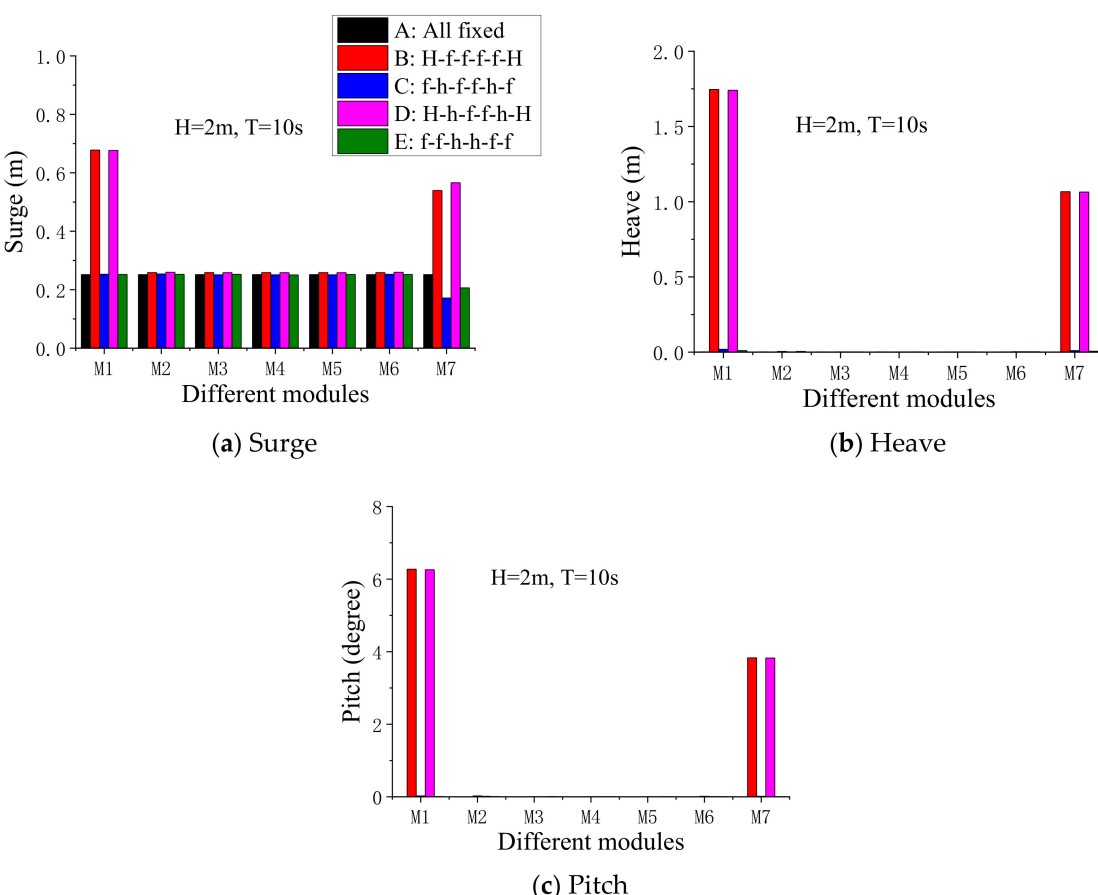

**Figure 5.** Main motion responses of the seven-module MTLPW system with different connection modes: (**a**) Surge, (**b**) heave, (**c**) pitch.

**Table 2.** Five connection modes for the MTLPW system.

| Case Name | Connection Mode |
|-----------|-----------------|
| A | All fixed |
| B | H-f-f-f-f-H |
| C | f-h-f-f-h-f |
| D | H-h-f-f-h-H |
| E | f-f-h-h-f-f |

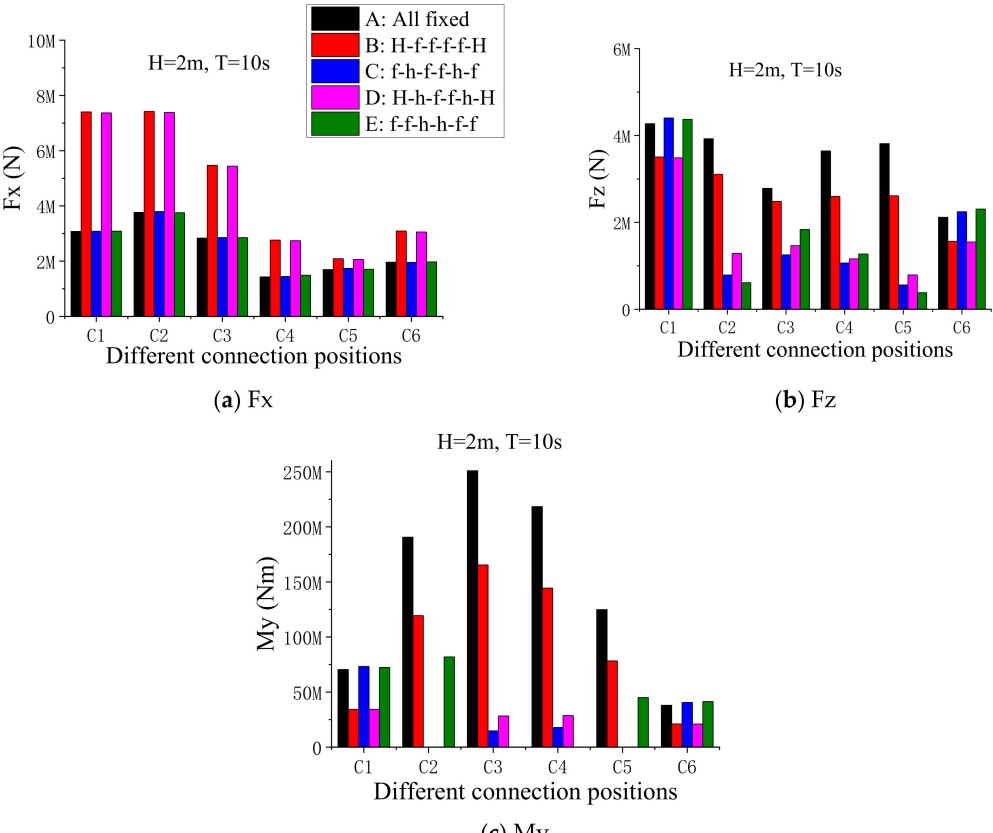

**Figure 6.** Connector loads of the seven-module MTLPW system with different connection modes: (**a**) Fx, (**b**) Fz, (**c**) My.

In Figure 5a–c, the two outermost connectors for the WEC modules (C1 and C6) had more significant influence on the motion responses of the proposed system than the connectors among TLP modules (C2~C5). Compared with the outermost HWK connector (C1 and C6), the outermost fixed connector effectively reduced the motion responses of the WEC modules. In addition, there was no bottom impact force observed for all the five connection modes.

In Figure 6a, the maximum horizontal connection force (Fx) occurred at the up-wave C1 or C2. The Fx amplitudes of all connectors were sensitive to the connector type of the outermost connector C1 (C6), and they were significantly reduced by the outermost fixed connector. In Figure 6b, it can be seen that the maximum vertical connection force (Fz) always occurred at the outermost connector C1, and it was effectively mitigated by the outermost HWK connector. Comparing Figure 6a with Figure 6b, it is obvious that the maximum response of Fz was much smaller than that of Fx. This is because tension legs can well limit relative heave responses among adjacent TLP modules, which tends to result in smaller vertical connection loads.

In Figure 6c, the outermost HWK or C2~C5 hinge connector types significantly reduced the amplitude of the connectors' bending moment (My). The maximum My usually

occurs in the middle of the continuously fixed connectors, and it tends to considerably increase as the number of the continuously fixed modules increases. Therefore, the number of the continuously fixed modules should be properly limited to control the extreme My responses of the proposed system.

From Figures 5 and 6, it can be concluded that the hydrodynamic responses for the connection modes Case C and Case D seemed much better than the other three connection modes, especially for the challenging My responses. In addition, it may be convenient to switch one connection mode (Case C) to another one (Case D) by only changing the outermost connector type. Case D can be applied for producing wave power in mild sea cases, as well as reducing the My responses on the outermost connectors. Case C can be used to limit the extreme motion responses of the outermost WEC modules to avoid possible dangerous bottom impact accidents during extreme sea cases. Therefore, the two suggested connection modes (Case C and Case D) are mainly investigated in the following sections.

### 3.2. WEC Performance for Case D

In order to preliminarily optimize the performance of the WEC module for Case D, the PTO damping effect of the outermost HWK connectors on the WEC performance was studied under a selected sea case ($H$ = 2 m, $T$ = 10 s). The main results are shown in Figure 7. In Figure 7a, the My loads of the down-wave C6 were much smaller than those of the up-wave C1 due to the "shadow effect" of the inner modules. The My responses of the outermost connector tended to increase as the PTO damping (Kp) increased. In Figure 7b, it can be concluded that the optimal PTO damping was about $5.0 \times 10^8$ Nms/rad for the most power production. In addition, the main motion responses of the WEC modules are shown in Figure 7c. The responses of both the pitch and the heave decreased as the PTO damping increased, and no bottom impact forces among modules were observed. For practical sea cases, the optimal design of the PTO damping should balance both the power production and the extreme motion responses of the WEC modules to avoid possible bottom impact accidents.

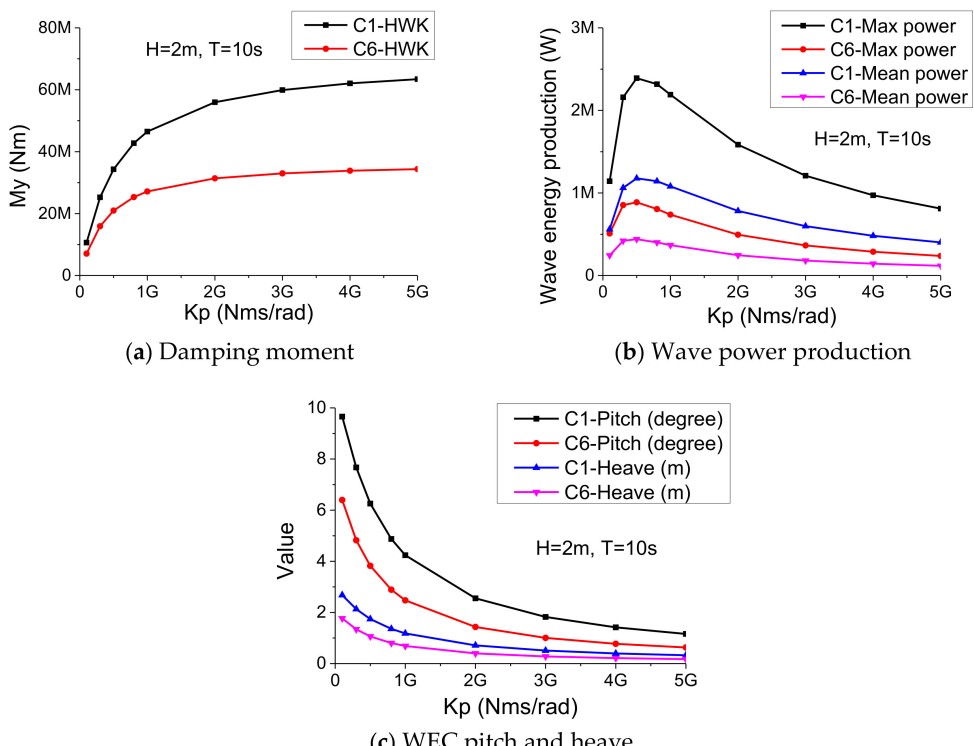

**Figure 7.** Effect of different pitch damping levels on WEC modules' motions: (**a**) Damping moment, (**b**) wave power production, (**c**) WEC pitch and heave.

The performance of the WEC modules with the optimal PTO damping ($Kp = 5.0 \times 10^8$ Nms/rad) under different wave periods is shown in Figure 8. The amplitudes of both the connectors' My and the power production of the WEC modules reached the peak for the wave period of around 10 s, which is consistent with the result in Figure 4. Therefore, the optimal wave period for the wave power production of the seven-module MTLPW system (with PTO damping $Kp = 5.0 \times 10^8$ Nms/rad) is about 10 s, while it is slightly different from that of the three-module MTLPW system (12 s) [29]. It can be concluded that, to some degree, the optimal wave period for wave power production of different MTLPW systems can be influenced by the number of inner TLP modules due to different hydrodynamic interaction effects.

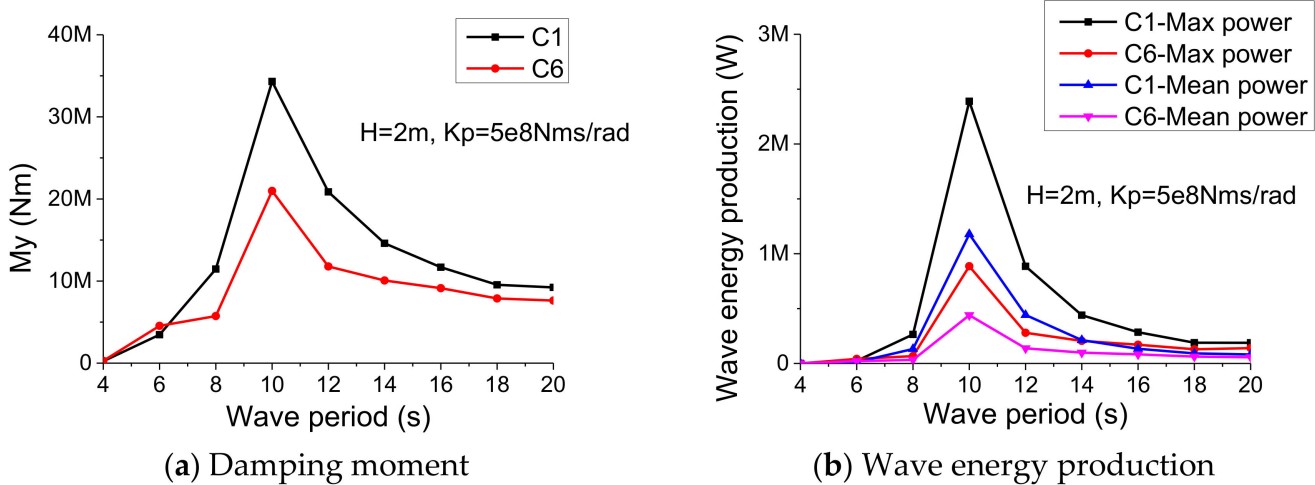

(**a**) Damping moment  (**b**) Wave energy production

**Figure 8.** Performance of the WEC modules under different wave periods: (**a**) Damping moment, (**b**) wave energy production.

### 3.3. Comparison of Two Suggested Connection Modes

The seven-module MTLPW system with two suggested connection modes of Case C and Case D were further investigated and compared under different wave periods. Considering that the two outermost connectors (C1 and C6) are symmetrically distributed and the up-wave C1 tended to suffer larger wave loads without the modules' "shadow effect", the connection loads on C1 were mainly focused, as shown in Figure 9. For Case C, the Fx amplitude reached the peak at the wave period of 6 s, while the amplitudes of Fz and My steadily increased as the wave period increased. For Case D, the maximum amplitudes of the Fx, Fz and My all occurred when the wave period of 10 s, and they tended to decrease when the wave period was larger than 10 s. Compared with Case C, Case D effectively reduced the amplitudes of both the Fz and the My, especially when the wave period was over 10 s.

In addition, the connector loads of the C2~C5 among the TLP modules for the two suggested connection modes are shown in Figure 10. All the maximum amplitudes of Fx, Fz and My for the two suggested connection modes were not significantly different. Therefore, the connector loads of the inner TLP connectors were not sensitive to the change of the connection mode between Case C and Case D.

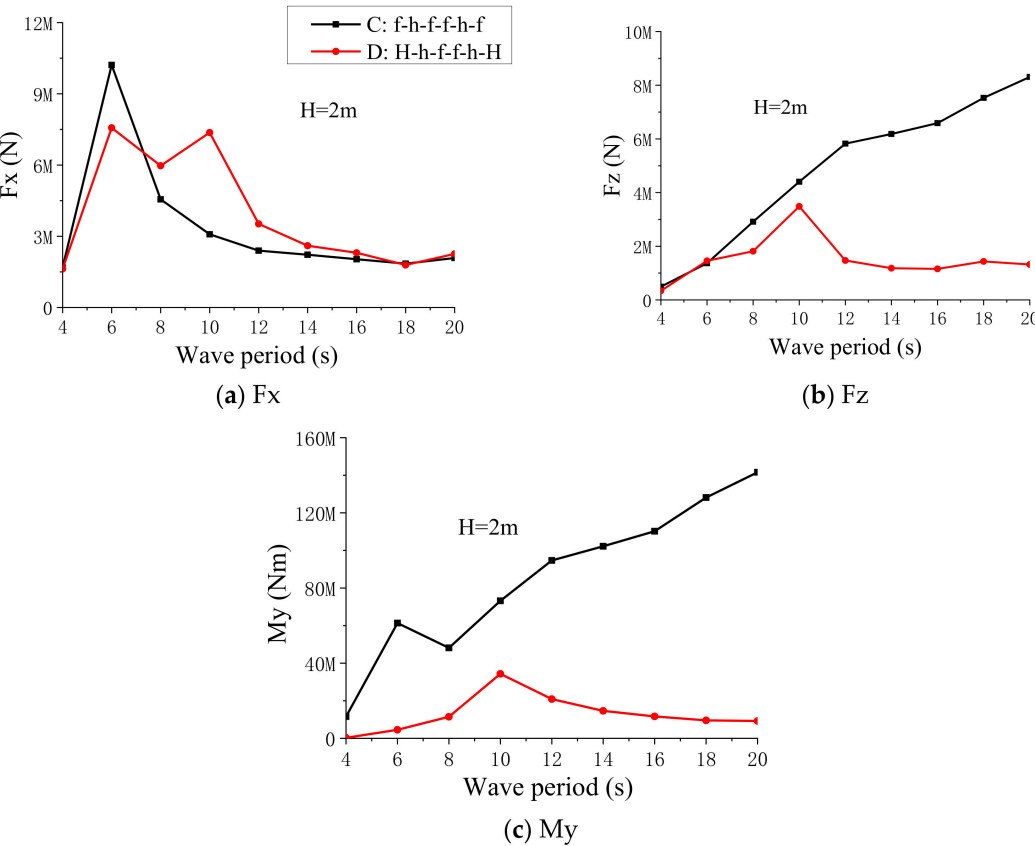

**Figure 9.** Main loads of the C1 with two suggested connection modes under different wave periods: (**a**) Fx, (**b**) Fz, (**c**) My.

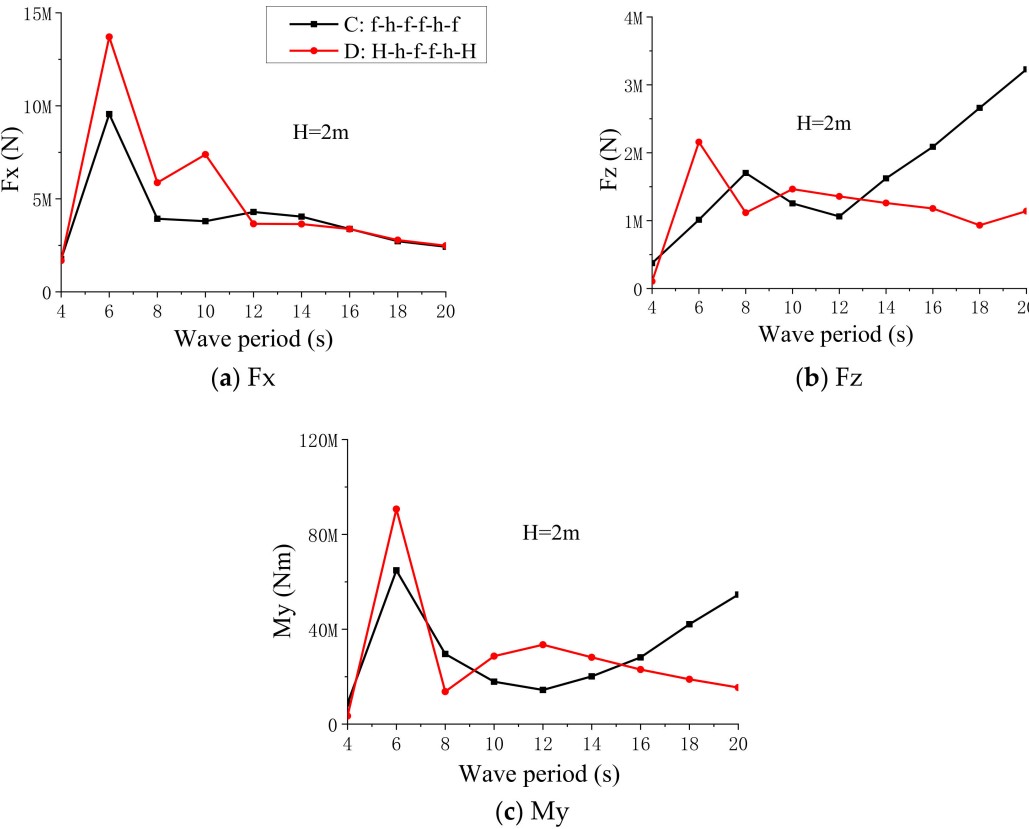

**Figure 10.** Maximum connector loads with two suggested connection modes under different wave periods: (**a**) Fx, (**b**) Fz, (**c**) My.

### 3.4. The Number of the Inner TLP Modules

To further study the possibility of expanding the MTLPW system, the effect of the number of the inner TLP modules on the hydrodynamic responses of the proposed system was investigated. Four typical modular systems were taken into consideration: The three-module (H-H) system, the five-module (H-f-f-H) system, and the seven-module system for Case B, and the seven-module system for Case D. The outermost connectors for the two WEC modules of the four selected modular systems were all the HWK type, with a damping of $5.0 \times 10^8$ Nms/rad. The comparison of main motion responses of the four selected systems is shown in Figure 11. It should be pointed out that, for each system, the M1 and the M7 indicate the up-wave and down-wave WEC modules, respectively, and the M2~M6 indicate the inner TLP modules. From Figure 11a–c, it can be concluded that the motion responses of the MTLPW system were not very sensitive to the number of TLP modules, and no bottom impact accident was observed.

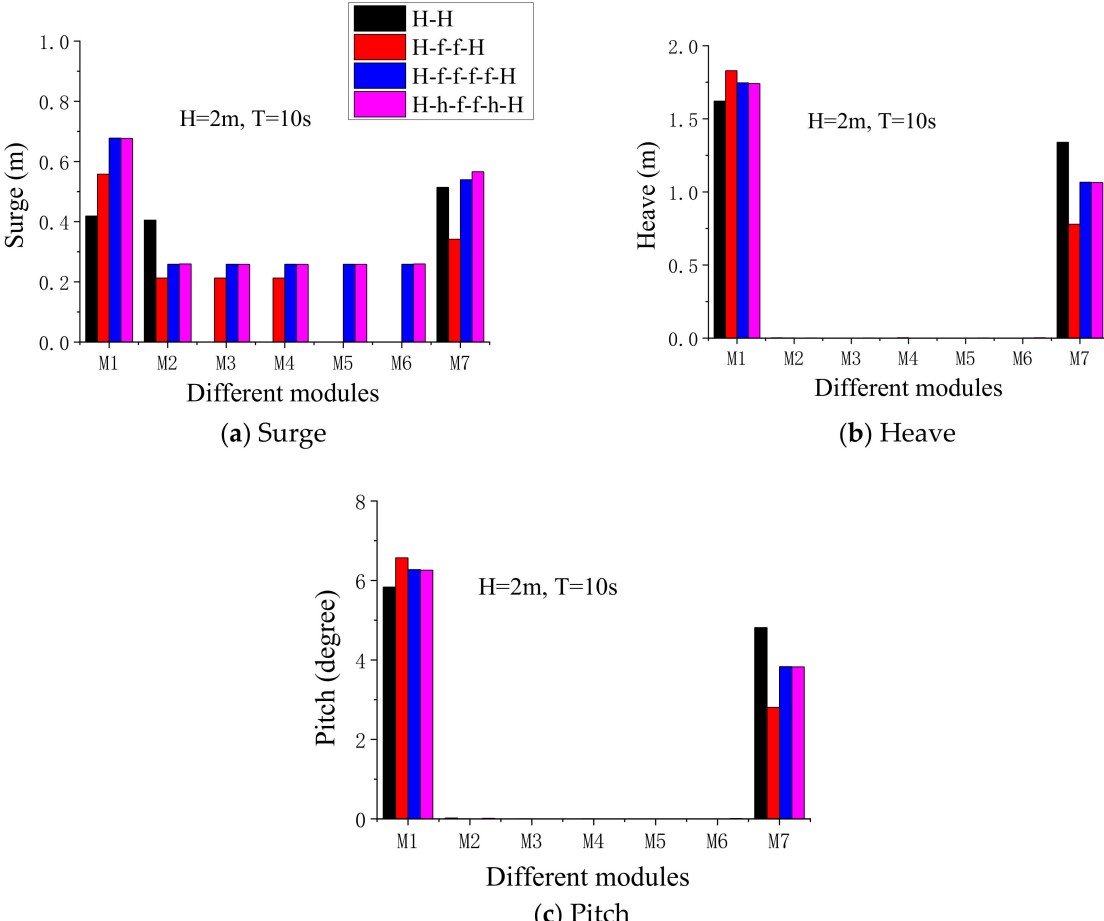

**Figure 11.** Comparison of main motion responses of the MTLPW system with different number of inner tension-leg platforms (TLP) modules: (**a**) Surge, (**b**) heave, (**c**) pitch.

The connector loads of the four selected systems were also investigated, as shown in Figure 12. It should be noted that, for each system, the C1 and the C6 indicate the connectors for the up-wave and down-wave WEC modules, respectively, and the C2~C5 indicate the connectors among the TLP modules. In Figure 12a,b, the maximum responses of both Fx and Fz gradually increased as the number of TLP modules increased. However, in Figure 12c, as the number of the TLP modules increased, the maximum responses of the My significantly increased, especially for the inner TLP modules with the fixed connectors (about four times larger than that of three-module system). However, the

maximum My response of the seven-module system for Case D was even less than that of the three-module system.

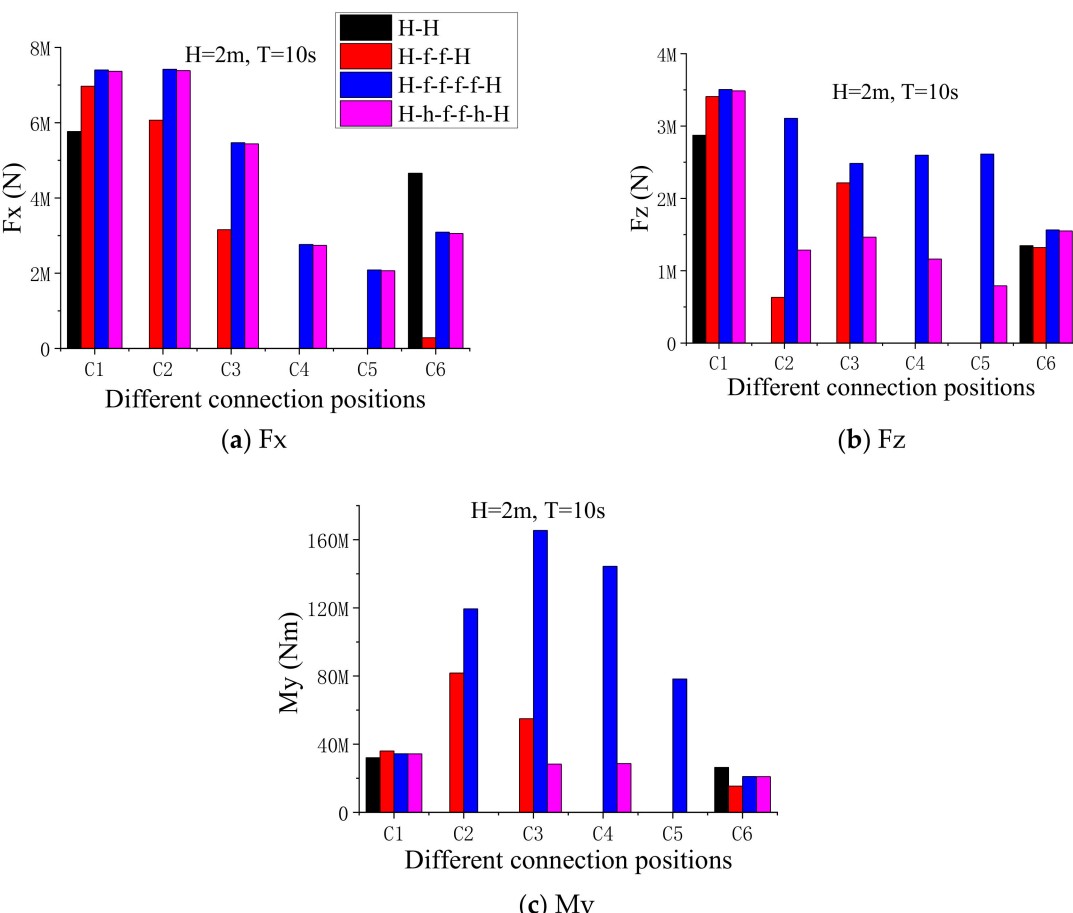

**Figure 12.** Comparison of main connectors' loads of the MTLPW system with different number of inner TLP modules: (**a**) Fx, (**b**) Fz, (**c**) My.

Therefore, the influence of the more inner TLP modules on the maximum My responses of connectors can be effectively mitigated by properly using hinge connectors to limit the number of the continuously fixed modules, meaning that the MTLPW can be potentially expanded to a large degree.

### 3.5. Extreme Sea Conditions

One typical mild sea zone with a natural island shield effect was selected for the investigation of the extreme responses of the seven-module MTLPW system (mainly concerning the two suggested connection modes). Considering that the practical sea is always complex with random features, one representative extreme irregular wave condition was considered for the JONSWAP( Joint North Sea Wave Project) spectrum ($\gamma = 3.3$, $Hs = 4$ m, $Tp = 10$ s) [32]. Main statistical information of extreme hydrodynamic responses of the seven-module MTLPW system under the selected sea zone is shown in Table 3.

In Table 3, for Case D (with the PTO damping of the outermost connectors of $5.0 \times 10^8$ Nms/rad), the maximum transient wave energy power reached 26 MW, but the outermost WEC modules tended to suffer significantly large pitch and heave extreme responses (over 10° and 3 m, respectively). Large relative pitch motions between the outermost WEC module and its adjacent TLP module led to their bottom impact accidents (observed over 30 times per hour), and the extreme impact force can reach 2.0 MN. Therefore, the preliminary PTO damping ($5.0 \times 10^8$ Nms/rad) may not have been large enough to successfully limit the extreme relative pitch motions between the outermost WEC mod-

ule and its adjacent TLP module, so a higher PTO damping ($Kp = 2.0 \times 10^9$ Nms/rad) was proposed for the extreme sea condition. In Table 3, compared with Case D, the higher PTO damping (Case D*) tended to increase the connectors' extreme Fz and My responses to some degree, but it significantly reduced the WEC modules' extreme pitch and heave responses by 48% and 47%, respectively. It is very beneficial for avoiding terrible bottom impact accidents among modules. It should be pointed out that there was no bottom impact force observed for Case D*. In addition, the mean wave energy production of the combined system of Case D* considerably reached 2 MW, although it was slightly smaller than that of Case D. Compared with Case D and Case D*, Case C had one of the smallest WEC modules' extreme pitch and heave responses, but it significantly induced the largest connectors' (Fz and My0 extreme responses, which is very challenging for the long-term safety design of the proposed system. Therefore, to balance extreme motion responses of the outermost WEC modules (avoiding terrible bottom impact accidents), extreme connector loads, and considerable wave energy production, the Case D* connection mode (with proper higher PTO damping) may be promising for practical extreme sea conditions.

**Table 3.** Statistical information of main extreme responses of the seven-module MTLPW system.

| Parameter Connector Mode | Fx (MN) | Fz (MN) | My (MNm) | WEC Pitch (Degree) | WEC Heave (m) | WEC Power (MW) |
|---|---|---|---|---|---|---|
| **Case C** | | | | | | |
| Max | 12.6 (C2) | 13.6(C1) | 224.0 (C1) | 0.08 | 0.06 | —— |
| Mean | 2.87 | 2.82 | 48.2 | 0.02 | —— | —— |
| STD | 2.17 | 2.19 | 37.2 | 0.01 | 0.01 | —— |
| **Case D** | | | | | | |
| $Kp = 5 \times 10^8$ Nms/rad | | | | | | |
| Max | 20.5 (C1) | 9.35 (C1) | 125.0 (C3) | 14.54 | 3.79 | 26.8 |
| Mean | 4.70 | 1.78 | 25.2 | 3.46 | —— | 2.30 |
| STD | 3.51 | 1.42 | 19.4 | 2.65 | 1.21 | 1.62 |
| **Case D*** | | | | | | |
| $Kp = 2 \times 10^9$ Nms/rad | | | | | | |
| Max | 16.0 (C1) | 10.1 (C1) | 150.0 (C1) | 7.53 | 2.01 | 22.4 |
| Mean | 3.66 | 2.21 | 35.3e7 | 1.65 | —— | 1.99 |
| STD | 2.78 | 1.70 | 27.1e7 | 1.27 | 0.58 | 1.41 |

## 4. Conclusions

This paper proposes a modular floating structure with inner tension-leg platforms and outermost wave energy converters. Both the mechanical coupling effect of connectors and the hydrodynamic interaction effect among modules were considered for the numerical model of a simplified seven-module MTLPW system. The effects of different connection modes, PTO damping levels of WEC modules, and the number of the inner TLP modules on the hydrodynamic characteristics of the proposed MTLPW system were clarified. The main numerical results can be summarized as follows:

(1) For the seven-module MTLPW system, the amplitudes of the connectors' Fx and Fz were more sensitive to the type of the outermost connectors than that of the inner connectors. The outermost HWK connector can effectively reduce the amplitudes of the connectors' Fz and My and produce considerable wave power. The outermost fixed connector is effective for reducing the amplitudes of the connectors' Fx and the motion responses of the WEC modules, as well as avoiding possible bottom collision accidents.

(2) Two suggested connection modes were proposed for the seven-module MTLPW system, which refer to Case C (f-h-f-f-h-f) and Case D (H-h-f-f-h-H). That is because they can effectively mitigate main hydrodynamic responses of the seven-module MTLPW system, especially for the connectors' My. For Case D, the effect of the PTO damping level of the outermost HWK connector on the hydrodynamic performance

of the seven-module MTLPW system was also clarified. The preliminary optimal PTO damping and corresponding optimal wave period for the outermost HWK connector were about $5.0 \times 10^8$ Nms/rad and 10 s, respectively.

(3)     The module's motion responses and the connectors' Fx and Fz loads for the seven-module MTLPW system were not very sensitive to more inner TLP modules, while the connectors' My responses were significantly sensitive to the number of the continuously fixed modules. Maximum connectors' My responses can be effectively limited by replacing certain fixed connectors with hinge connectors to reduce the number of the continuously fixed modules. Therefore, the number of the inner TLP modules of the MTLPW system can be flexibly expanded with the proper hinge connector strategy.

(4)     For the outermost HWK connector mode (Case D) under practical extreme sea conditions, the bottom impact accidents between the outermost WEC module and its adjacent TLP module may occur. However, the bottom impact accidents can be effectively avoided by properly enlarged the PTO damping level to reduce the relative pitch motion among modules. To balance the long-term safety and the wave power production of the proposed MTLPW system, the Case D* connection mode (with proper higher PTO damping) may be a promising for practical extreme sea conditions.

Challenges for the proposed MTLPW system still remain, and the study of a more robust concept requires more efforts. It may include modeling a larger MTLPW system (involving more modules), scale model tests for the validation of the numerical model, and corresponding long-term performance analysis.

**Author Contributions:** Conceptualization, N.R. and J.O.; methodology, K.L. and D.Z.; formal analysis, H.W.; investigation, N.R. and K.L. All authors have read and agreed to the published version of the manuscript.

**Funding:** This research was supported by the Natural Science Foundation of Hainan Province (No. 520RC552, 520RC543), the National Natural Science Foundation of China (Grant No. 51709040) and the Science Foundation of Hainan University. The financial supports are greatly acknowledged.

**Institutional Review Board Statement:** The study was conducted according to the guidelines of the Declaration of Helsinki, and approved by the Institutional Review Board.

**Informed Consent Statement:** Informed consent was obtained from all subjects involved in the study.

**Conflicts of Interest:** The authors declare no conflict of interest.

## Abbreviations

| | |
|---|---|
| Ci | The i-th Connector |
| DOF | Degree of Freedom |
| FEA | Finite Element Analysis |
| HWK | Hinge connector with an additional WEC linear pitch damper (Kp) |
| JONSWAP | Joint North Sea Wave Project |
| Mi | The i-th Module |
| MOB | Mobile Offshore Base |
| MTLPW | Modular Tension Leg Platforms with Wave energy converters |
| OWC | Oscillating Water Column |
| PTO | Power Take-Off |
| RAO | Response Amplitude Operator |
| RMFC | Rigid-Module-Flexible-Connector |
| TLP | Tension Leg Platform |
| WEC | Wave Energy Converters |
| VLFS | Very Large Floating System |

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
