# Peer review of "Hydrodynamic Analysis of a Modular Floating Structure with Tension-Leg Platforms and Wave Energy Converters"

_jmse, doi:10.3390/jmse9040424_

Round 1
Reviewer 1 Report
GENERAL COMMENTS
The work is very interesting, it has a clear degree of originality, and it might be appropriate for publication in the journal, after performing a major and very careful revision.
Indeed the topic targeted is very important since marine renewable energy has a huge potential and thus represents an issue of increasing importance, especially in island environment. However, the proposed work should be still completed and improved in various points in order to be ready for a journal publication.
My main concern relates however the efficiency of the WECs considered for the structure presented in Figure 1. A power matrix should be provided for the system as well as LCOE projections, although indeed the hydro-elastic analysis should focused on the safety design of a very large floating structure.
Furthermore, more literature revue and discussions about similar research should be made and also further discussions and explanations related to the validity of the proposed work should be provided..
Please note also that the paper should be formatted according to the journal template.
Some specific comments are given next. They are not exhaustive, which means that there might be also still some other issues to be double checked by the authors before resubmission.
SPECIFIC COMMENTS
ENGLISH LANGUAGE
This is in general OK. However, an additional English check should be performed since there are still some sentences that need to be reformulated in order to present in a clearer way the ideas and the findings of the proposed work.
SYMBOLS AND EQUATIONS
There some are equations presented in the paper and most of them look OK. However please check carefully whether all the quantities involved are properly explained.
For example in equation (1) you should explained each generalized wave force matrix Fi,wave, Fi,con, Fi,tlp and Fi,fender.
Please check also equation (4), it seems that you have an unclear bracket.
The quantities from equation (5) are not at all explained (Bpto) and also some more details related to these computations should be provided.
ABBREVIATIONS
You have used many abbreviations in the text. From this perspective, an Index of Notations and Abbreviations would be beneficial for a better understanding of the proposed work.
Furthermore, please check carefully if all the abbreviations and notations considered in the work are explained for the first time when they are used, even if these are considered trivial by the authors. The paper should be accessible to a wide audience. For example even the significance of the wave converter MTLPW is not at all explained in the text instead the explication provided modular floating structure, which is consisted of five inner tension-leg platforms and two outermost wave energy converters does not fit with the acronym MTLPW. The same for HWK.
PAPER STRUCTURE
You should better include the las section (5 Future work), which is very short, into the section of Conclusions
FIGURES & TABLES
Some corrections are required in relationship with the figures and the figures captions, as follows:
Figure 4- please provide a colour version for this figure, since this is an open access journal colour figures will be much better;
Figures 5, 6, 11, 12 please explicit in the figure captions the significance of each subplot (a, b and c). Furthermore since the legend is identical for all subplots it would be better to provide one single legend for each figure.
Author Response
Dear editors and reviewers:
Thank you very much for the reviewer’s comments concerning our manuscript. Those comments are all valuable and very helpful for revising and improving our paper. We have studied comments carefully and have made correction which we hope could meet with approval. Revised portion are marked in red in the paper. The main revision in the paper and the responses to the reviewer’s comments are listed as follows:
Responses to the comments of reviewer #1:
The work is very interesting, it has a clear degree of originality, and it might be appropriate for publication in the journal, after performing a major and very careful revision.
Indeed the topic targeted is very important since marine renewable energy has a huge potential and thus represents an issue of increasing importance, especially in island environment. However, the proposed work should be still completed and improved in various points in order to be ready for a journal publication.
Answer: Thanks for this positive comment on our paper!
My main concern relates however the efficiency of the WECs considered for the structure presented in Figure 1. A power matrix should be provided for the system as well as LCOE projections, although indeed the hydro-elastic analysis should focused on the safety design of a very large floating structure.
Answer: Thanks for this good comment!
Just as the reviewer said, the power efficiency of the WEC is a key factor for the optimization of the modular system. Considering the detailed wave power performance of a similar three-module system has been well clarified in the following reference [32] (our previous work in 2019), here, only a brief description has been presented in the section 3.2 for the main power performance of the seven-module system in this paper. Therefore, this paper is mainly focus on the effect of the outermost WEC module on the hydro-elastic dynamic responses of the seven-module system.
[32] Ren N. X., Wu H. B., Ma Z., Ou J. P. Hydrodynamic analysis of a novel modular floating structure system with central tension-leg platforms [J]. Ships and Offshore Structures. 2019,
Furthermore, more literature revue and discussions about similar research should be made and also further discussions and explanations related to the validity of the proposed work should be provided.
Answer: Thanks for this good comment!
More related references about the research on modular floating structures have been added in the Introduction section. In addition, we are currently planning a scale model test of the seven-module system to validate our proposed work (as presented in Section 5 Future Work).
“If the outermost flexible connectors can be equipped with WEC power take-off (PTO) dampers, not only the motion responses of the outermost modules can be effectively reduced, but also considerable wave energy power can be obtained [28-29]. In addition, some researchers [30-31] have tried to combine floating offshore wind turbines with Oscillating Water Column (OWC) devices, and some promising findings of the new hybrid wind-wave energy systems have been pointed out. ”
[29]Zhang H. C., Xu D. L., Ding R., et al. Embedded Power Take-Off in hinged modularized floating platform for wave energy harvesting and pitch motion suppression. Renewable Energy, 2019, 138: 1176-1188.
[30]Mazarakos T., Konispoliatis D., Katsaounis G., et al. Numerical and experimental studies of a multi-purpose floating TLP structure for combined wind and wave energy exploitation, Mediterranean Marine Science, 2019, 20: 745-763,
[31]Michele S., Renzi E., Perez-Collazo C., et al. Power extraction in regular and random waves from an OWC in hybrid wind-wave energy systems, Ocean Engineering, 2019, 191: 106519.
[36] Bachynski E. E. Design and dynamic analysis of tension leg platform wind turbines. PhD thesis, Norwegian University of Technology, 2014.
Please note also that the paper should be formatted according to the journal template.
Answer: Thanks for this good comment!
The paper will be further formatted according to the journal template.
Some specific comments are given next. They are not exhaustive, which means that there might be also still some other issues to be double checked by the authors before resubmission.
Answer: Thanks for this kind suggestion!
The paper has been further carefully checked.
SPECIFIC COMMENTS
ENGLISH LANGUAGE
This is in general OK. However, an additional English check should be performed since there are still some sentences that need to be reformulated in order to present in a clearer way the ideas and the findings of the proposed work.
Answer: Thanks for this kind suggestion!
The paper has been further polished.
SYMBOLS AND EQUATIONS
There some are equations presented in the paper and most of them look OK. However please check carefully whether all the quantities involved are properly explained.
For example in equation (1) you should explained each generalized wave force matrix Fi,wave, Fi,con, Fi,tlp and Fi,fender.
Answer: Thanks for this kind suggestion!
The detailed description of the mentioned force matrix (Fi,wave, Fi,con, Fi,tlp and Fi, fender) has been added in section 2.
The seven modules are assumed to stay in harmonic motion excited by wave forces, so the total wave force on the i-th module can be calculated as follows:
(4)
In addition, the total tension leg force on the i-th module can be calculated as follows:
(5)
where Ɛij is the strain matrix of the j-th tension leg of the i-th module. Ei and Ai are the elasticity modulus and the section area of the tension leg for the i-th module, respectively.
The connector forces (Fi,con) acting on the i-th module induced by adjacent modules can be expressed as:
(5)
where φij is a topology matrix. The φij is set to be one when the j-th module connects with the i-th module, otherwise φij is set to be zero. Kcij is the connection stiffness matrix between the i-th module and the j-th module. δ(Xi, Xj) is the relative motion matrix between the i-th module and the j-th module.
In addition, the possible bottom fender impact force Fi,fender can be simplified estimated as follows:
(6)
where Kfij (1.0×107 N/m) is the bottom fender stiffness coefficient between the i-th module and the adjacent j-th module. δx(Xi, Xj) is the relative bottom surge motion between the i-th module and the adjacent j-th module. If the negative relative bottom surge motion δx(Xi, Xj) is small than the module’s gap (2 m), the two adjacent modules will be bottom contact and the bottom fender contact force will be observed.
Please check also equation (4), it seems that you have an unclear bracket.
Answer: Thanks for this kind suggestion!
The brackets in equation (4) have been corrected.
(4)
The quantities from equation (5) are not at all explained (Bpto) and also some more details related to these computations should be provided.
Answer: Thanks for these good comment!
In fact, the parameter Bpto is the same as the parameter Kp, so the equation (5) has been corrected and updated.
(8)
ABBREVIATIONS
You have used many abbreviations in the text. From this perspective, an Index of Notations and Abbreviations would be beneficial for a better understanding of the proposed work.
Furthermore, please check carefully if all the abbreviations and notations considered in the work are explained for the first time when they are used, even if these are considered trivial by the authors. The paper should be accessible to a wide audience. For example even the significance of the wave converter MTLPW is not at all explained in the text instead the explication provided modular floating structure, which is consisted of five inner tension-leg platforms and two outermost wave energy converters does not fit with the acronym MTLPW. The same for HWK.
Answer: Thanks for this kind suggestion!
An additional ABBREVIATIONS section has been added in this paper.
|
MTLPW |
Modular Tension Leg Platforms with Wave energy converters |
|
TLP |
Tension Leg Platform |
|
WEC |
Wave Energy Converters |
|
PTO |
Power Take-Off |
|
VLFS |
Very Large Floating System |
|
RMFC |
Rigid-Module-Flexible-Connector |
|
FEA |
Finite Element Analysis |
|
MOB |
Mobile Offshore Base |
|
HWK |
Hinge connector with an additional WEC linear pitch damper (Kp) |
|
Ci |
The i-th Connector |
|
Mi |
The i-th Module |
In addition, the full-size of the proposed MTLPW system is shown in Fig.1. Considering the paper mainly focuses on the preliminary feasibility study of the MTLPW system, to balance the accuracy of the numerical model and corresponding computational time, a simplified seven-module MTLPW system is proposed for the investigation of its hydrodynamic characteristics in a mild sea zone.
PAPER STRUCTURE
You should better include the last section (5 Future work), which is very short, into the section of Conclusions
Answer: Thanks for this kind suggestion!
Considering the last section (Future work) is our coming plan (not done yet), it may be possible to set the “Future work” as an independent section due to the difference from the already done work in the Conclusion section.
FIGURES & TABLES
Some corrections are required in relationship with the figures and the figures captions, as follows:
Figure 4- please provide a colour version for this figure, since this is an open access journal colour figures will be much better;
Figures 5, 6, 11, 12 please explicit in the figure captions the significance of each subplot (a, b and c). Furthermore since the legend is identical for all subplots it would be better to provide one single legend for each figure.
Answer: Thanks for these kind suggestion!
Considering only one curve in Fig.4, the color of the curve has been set as black.
The mentioned figures have been updated according to the suggestion.
Special thanks for your valuable time and wonderful comments.
Yours sincerely,
Best regards,
Nianxin Ren

Reviewer 2 Report
Abstract
Typical sea conditions. That means operating or survival or regular wave analysis? Please give more details.
Introduction
Line 6: military base: give some references
2.1 Description of the MTLPW
… Characteristics in a mild sea zone… Give more details – references.
Table 1: The t=0.04m in naval architecture is very small. Give more details for the material of the Tension Leg.
Give the restoring forces of the TLP.
Give the pretension used in this study.
Stiffness of bottom fenders: give more details for the fenders
WEC PTO damping: more details for the damping values
… Each TLP module is initially designed to withstand about 2000t mass variation,… This phrase it isn’t clear, give more details.
2.2 Governing equation
This chapter needs more details. Please give the Mi, Ci an Ki matrixes respectively used in this study.
How you calculate the Ci? Due to the hydrodynamic analysis? More details – references are needed.
2.3 Hydrodynamic model
This chapter needs more details.
Equation (3): Please re-write the equation. Some parenthesis are missing.
Also more details are needed i.e. please give the definition of ω, ρ, ni
Equation (4): something is missing in equation (4). Give the definition of U
Equations for the RAO’s?
2.4 Connector types
Give more details for the damping. How you select this value in your study?
2.5 Estimation of wave power production
This chapter need more details- references. ANSYS –AQWA calculates the wave power? Give a reference on that.
3.1 Effects of different connection modes
Fig. 4: Due to the design of floating structures 25 degrees is a high value. Is this value acceptable? the tendons can withstand this angle? The structure is stable? Have you check the stability?
3.2 WEC performance for the Case D
Give more details for the PTO damping. Why the optimal damping is about 5*10*?
3.5 Extreme sea conditions
For the Jonswap spectrum please write γ=3.3. Give the equation describes the Jonswap spectrum used in this study. How you choose the values of the extreme irregular sea conditions. Give a characteristic area with this sea conditions.
Table 3: more statistics are needed (min, stdev, mean). Give an explanation for the high values of the Heave (WEC): 3.79, 2.01m
General comments
For the Introduction
I believe that the description of the current state-of-art on wave energy converters, in general, and on the OWC devices, in the specific, should be enhanced. Some works on the OWC-WECs miss in the Introduction. At this regard, please consider the inclusion of the works by Mazarakos et al. 2019 (Numerical and experimental studies of a multi-purpose floating TLP structure for combined wind and wave energy exploitation, Mediterranean Marine Science, 20|2019, 745-763), S. Michele, et al., 2019 (Power extraction in regular and random waves from an OWC in hybrid wind-wave energy systems, Ocean Engineering, Volume 191, 2019, 106519, ISSN 0029-8018).
More details for the model design are needed (i.e. number of diffracted elements).
Details for the TLP system used and more details for the theoretical background used for the joint, hinge, fender analysis and calculation.
Author Response
Dear editors and reviewers:
Thank you very much for the reviewer’s comments concerning our manuscript. Those comments are all valuable and very helpful for revising and improving our paper. We have studied comments carefully and have made correction which we hope could meet with approval. Revised portion are marked in red in the paper. The main revision in the paper and the responses to the reviewer’s comments are listed as follows:
Responses to the comments of reviewer #2:
Abstract: Typical sea conditions. That means operating or survival or regular wave analysis? Please give more details.
Answer: Thanks for this good comment!
The description of typical sea conditions in Abstract section has been clarified as “both operational and extreme sea conditions”.
Introduction: Line 6: military base: give some references
Answer: Thanks for this good comment!
Considering there are very few references about the military base, the corresponding introduction of the military base has been deleted.
2.1 Description of the MTLPW
Characteristics in a mild sea zone… Give more details – references.
Answer: Thanks for this good comment!
Considering the modular VLFS usually locals at certain sea zones with the protection of natural islands or artificial seawalls, so the sea conditions for the proposed MTLPW can refer to a certain mild sea zone in South China sea (reference [32]) .
[32] Ren N. X., Wu H. B., Ma Z., Ou J. P. Hydrodynamic analysis of a novel modular floating structure system with central tension-leg platforms [J]. Ships and Offshore Structures. 2019, https://doi.org/10.1080/17445302.2019.1700035
Table 1: The t=0.04m in naval architecture is very small. Give more details for the material of the Tension Leg.
Answer: Thanks for this good comment!
The dimension information of one single steel tension leg is shown in Table 1. Usually the self weight of the tension leg is equal to its buoyancy, so the thickness of the tension leg with the diameter of 1.2 m is about 0.04 m. More detail design information can refer to the reference [32] and the reference [36].
Give the restoring forces of the TLP.
Answer: Thanks for this good comment!
More detail restoring forces of the TLP information can refer to the reference [32] and the reference [36]. The corresponding references have been introduced for the design description of the inner TLP module.
Give the pretension used in this study.
Answer: Thanks for this good comment!
The total pretension information has been added in Table 1. It can be calculated by the total mass and total displacement information.
Stiffness of bottom fenders: give more details for the fenders
Answer: Thanks for this good comment!
More detailed information of the bottom fenders has been added in the section 2.4. (Shown as follows: )
The possible bottom fender impact force Fi,fender can be simplified estimated as follows:
(7)
where Kfij (1.0×107 N/m) is the bottom fender linear stiffness coefficient between the i-th module and the adjacent j-th module. δx(Xi, Xj) is the relative bottom surge motion between the i-th module and the adjacent j-th module. If the relative bottom surge motion δx(Xi, Xj) is small than the module’s gap (2 m), the two adjacent modules will be bottom contact and the bottom fender contact force will be observed.
WEC PTO damping: more details for the damping values
Answer: Thanks for this good comment!
The detailed information of the WEC PTO damping (Kp) are shown in the Section 3 (Numerical results). The optimal PTO damping (for the relative pitch motion) is about 5.0×108 Nms/rad for the most power production, while a higher PTO damping (Kp=2.0×109 Nms/rad) has been proposed for the extreme sea condition.
Each TLP module is initially designed to withstand about 2000t mass variation,… This phrase it isn’t clear, give more details.
Answer: Thanks for this good comment!
The corresponding description has been revised as follows for a better understanding.
“Each TLP module is initially designed to withstand about 2000t mass variation (available for TLP pretension range of 3000~5000 t)”
2.2 Governing equation
This chapter needs more details. Please give the Mi, Ci an Ki matrixes respectively used in this study.
How you calculate the Ci? Due to the hydrodynamic analysis? More details – references are needed.
Answer: Thanks for this good comment!
The AQWA code is available and widely used for the hydrodynamic analysis of the TLP, the multi-bodies’ hydrodynamic interaction effect, and the mechanical coupling effect, and more detailed information can refer to the reference [32] and the reference [33]. Therefore, only a brief description is presented in this paper.
2.3 Hydrodynamic model: This chapter needs more details.
Answer: Thanks for this good comment!
More detailed description of the hydrodynamic model has been added.
In addition, the total tension leg force on the i-th module can be calculated as follows:
(5)
where Ɛij is the strain matrix of the j-th tension leg of the i-th module i. Ei and Ai are the elasticity modulus and the section area of the tension leg of the i-th module.
Equation (3): Please re-write the equation. Some parenthesis are missing.
Answer: Thanks for this good comment!
The Equation (3) has been corrected.
Also more details are needed i.e. please give the definition of ω, ρ, ni
Equation (4): something is missing in equation (4). Give the definition of U
Equations for the RAO’s?
Answer: Thanks for these good comment!
The corresponding description has been revised.
2.4 Connector types
Give more details for the damping. How you select this value in your study?
Answer: Thanks for these good comment!
The detailed information of the WEC PTO damping (Kp) are shown in the Section 3 (Numerical results). The suggested optimal PTO damping is about 5.0×108 Nms/rad for the most power production, which are based on the numerical results in Fig.7b.
2.5 Estimation of wave power production
This chapter need more details-references. ANSYS –AQWA calculates the wave power? Give a reference on that.
Answer: Thanks for these good comment!
The AQWA can output the WEC connectors’ bending moment information of the pitch damper (Mbpto), then, the wave power information can be calculated by the Equation (8) (with a constant parameter Kp , such as 5.0×108 Nms/rad).
For the two outermost HWK connectors (C1 and C6), the WEC PTO systems have been simplified as linear pitch dampers (Kp). The bending moment of the pitch damper (Mbpto) and the corresponding relative pitch velocity (wref) between the WEC module and the adjacent TLP module can be used to estimate the power production of the WEC (Pwave). The corresponding formula can be written as follows:
(8)
3.1 Effects of different connection modes
Fig. 4: Due to the design of floating structures 25 degrees is a high value. Is this value acceptable? the tendons can withstand this angle? The structure is stable? Have you check the stability?
Answer: Thanks for these good comment!
Fig.4 shows the pitch RAO of a single free WEC module (without tension legs), so the pitch amplitude (for the wave period of about 10s) can almost reach 25 degrees. The information of the pitch RAO of the free WEC module is helpful for estimating the natural pitch period of the outermost free WEC module to optimal the WEC performance. However, the real pitch responses of the WEC module of the MTLPW system can’t reach such a high value (25 degree) due to the effect of the connectors between the outermost WEC module and its adjacent TLP module, which can be seen in Table 3.
3.2 WEC performance for the Case D
Give more details for the PTO damping. Why the optimal damping is about 5*10*?
Answer: Thanks for these good comment!
The detailed information of the WEC PTO damping (Kp) are shown in the Section 3 (Numerical results), especially for the section 3.2. The suggested optimal PTO damping is about 5.0×108 Nms/rad for the most power production, which are based on the numerical results in Fig.7b.
- b) Wave power production
Fig.7 Effect of different pitch damping levels on WEC modules’ motions
3.5 Extreme sea conditions
For the Jonswap spectrum please write γ=3.3. Give the equation describes the Jonswap spectrum used in this study. How you choose the values of the extreme irregular sea conditions. Give a characteristic area with this sea conditions.
Answer: Thanks for these good comment!
The corresponding description has been revised.
Considering the JONSWAP spectrum is widely used and can be easily find in most ocean engineering books, more detailed description of the equation has been omitted here. The extreme irregular sea conditions can refer to the corresponding reference [32] for more information.
Table 3: more statistics are needed (min, stdev, mean). Give an explanation for the high values of the Heave (WEC): 3.79, 2.01m
Answer: Thanks for these good comment!
Considering the dynamic responses’ absolute value of the “Min value” is smaller than the “Max value” to some degree (based on our numerical statistical data), so the “Min value” of the main extreme responses is omitted here.
For the Case D, the extreme heave response can reach 3.79 m due to too small connector’s pitch damper (5.0×108 Nms/rad), and the bottom impact force (2.0 MN) can also be observed. In other words, the case D is dangerous and may be unacceptable for safety design of the proposed MTLPW system. Therefore, the Case D* (with larger connector’s pitch damper of 2.0×109 Nms/rad)) has been proposed for improving the extreme responses of the MTLPW system. For the Case D*, the extreme heave response can be effectively reduced to 2.01 m, and the bottom impact accident can be successfully avoided. Considering the Hs is 4 m under the extreme sea condition, so the extreme heave response (2.01 m) of the WEC module may be acceptable without significant negative effects.
General comments: For the Introduction
I believe that the description of the current state-of-art on wave energy converters, in general, and on the OWC devices, in the specific, should be enhanced. Some works on the OWC-WECs miss in the Introduction. At this regard, please consider the inclusion of the works by Mazarakos et al. 2019 (Numerical and experimental studies of a multi-purpose floating TLP structure for combined wind and wave energy exploitation, Mediterranean Marine Science, 20|2019, 745-763), S. Michele, et al., 2019 (Power extraction in regular and random waves from an OWC in hybrid wind-wave energy systems, Ocean Engineering, Volume 191, 2019, 106519, ISSN 0029-8018).
Answer: Thanks for this good comment!
More references about the research on modular floating structures have been added in the Introduction section.
“If the outermost flexible connectors can be equipped with WEC power take-off (PTO) dampers, not only the motion responses of the outermost modules can be effectively reduced, but also considerable wave energy power can be obtained [28-29]. In addition, some researchers [30-31] have tried to combine floating offshore wind turbines with Oscillating Water Column (OWC) devices, and some promising findings of the new hybrid wind-wave energy systems have been pointed out. ”
[29]Zhang H. C., Xu D. L., Ding R., et al. Embedded Power Take-Off in hinged modularized floating platform for wave energy harvesting and pitch motion suppression. Renewable Energy, 2019, 138: 1176-1188.
[30]Mazarakos T., Konispoliatis D., Katsaounis G., et al. Numerical and experimental studies of a multi-purpose floating TLP structure for combined wind and wave energy exploitation, Mediterranean Marine Science, 2019, 20: 745-763,
[31]Michele S., Renzi E., Perez-Collazo C., et al. Power extraction in regular and random waves from an OWC in hybrid wind-wave energy systems, Ocean Engineering, 2019, 191: 106519.
More details for the model design are needed (i.e. number of diffracted elements).
Details for the TLP system used and more details for the theoretical background used for the joint, hinge, fender analysis and calculation.
Answer: Thanks for this kind suggestion!
The corresponding description has been revised, and it is suggested that more detailed information for the TLP system (the theoretical background used for the joint, hinge, fender analysis and calculation) can refer to our previous work (references [32-35]).
[32] Ren N. X., Wu H. B., Ma Z., Ou J. P. Hydrodynamic analysis of a novel modular floating structure system with central tension-leg platforms [J]. Ships and Offshore Structures. 2019, https://doi.org/10.1080/17445302.2019.1700035
[33]ANSYS, Inc. ANSYS AQWA User's Manual, 2010.
[34]Ren N. X., Li Y. G., Ou J. P. The wind-wave tunnel test of a tension-leg platform type floating offshore wind turbine. Journal of Renewable and Sustainable Energy[J]. 2012,4(6): 63117.
[35]Ren N. X., Li Y. G., Ou J. P. Coupled wind-wave time domain analysis of floating offshore wind turbine based on Computational Fluid Dynamics method. Journal of Renewable and Sustainable Energy[J]. 2014,6(2): 23106.
Special thanks for your valuable time and wonderful comments.
Yours sincerely,
Best regards,
Nianxin Ren

Round 2
Reviewer 1 Report
Most of my observations and suggestions have been considered and appropriate corrections have been operated.
There are however some minor issues that should be fixed before the paper being ready for journal publications.
- The paper is still not written according to the journal template;
- In the right hand of equation 3 you have an unclear bracket, please correct;
- Please use the alphabetical order in the Abbreviation list, furthermore please check carefully whether all the abbreviations are included in this list. For example DOF is not included.
- Figure 4 – please provide a color version also for this figure;
- Figures 7, 8 please explain also each subplot in the figures captions.
- Figures 5, 6, 9, 10, 11, 12 – since the legend is the same for all three subplots you could include it only once. Furthermore please explain also each subplot in the figures captions.
- You can include the last section (Future works) in the section of Conclusions
Author Response
Dear editors and reviewers:
Thank you very much for the reviewer’s further comments concerning our manuscript. We have studied comments carefully and have made correction which we hope could meet with approval. Revised portion are marked in red in the paper. The main revision in the paper and the responses to the reviewer’s comments are listed as follows:
Responses to the comments of reviewer #1:
Most of my observations and suggestions have been considered and appropriate corrections have been operated.
Answer: Thanks for this positive comment on our revised paper!
There are however some minor issues that should be fixed before the paper being ready for journal publications.
- The paper is still not written according to the journal template;
Answer: Thanks for this good comment!
The paper has been further formatted according to the journal template, especially for Figures and reference list.
- In the right hand of equation 3 you have an unclear bracket, please correct;
Answer: Thanks for this good comment!
The equation 3 has been corrected.
(3)
- Please use the alphabetical order in the Abbreviation list, furthermore please check carefully whether all the abbreviations are included in this list. For example DOF is not included.
Answer: Thanks for this good comment!
The Abbreviation list has been revised as follows:
|
Ci |
—The i-th Connector |
|
DOF |
—Degree of Freedom |
|
FEA |
—Finite Element Analysis |
|
HWK |
—Hinge connector with an additional WEC linear pitch damper (Kp) |
|
Mi |
—The i-th Module |
|
MOB |
—Mobile Offshore Base |
|
MTLPW |
—Modular Tension Leg Platforms with Wave energy converters |
|
OWC |
—Oscillating Water Column |
|
PTO |
—Power Take-Off |
|
RMFC |
—Rigid-Module-Flexible-Connector |
|
TLP |
—Tension Leg Platform |
|
WEC |
—Wave Energy Converters |
|
VLFS |
—Very Large Floating System |
- Figure 4 – please provide a color version also for this figure;
Answer: Thanks for this good comment!
The figure 4 has been in red color.
Fig.4 Pitch RAO of the free WEC module
- Figures 7, 8 please explain also each subplot in the figures captions.
Answer: Thanks for this good comment!
Each subplot has been added to the mentioned figures’ captions
- Figures 5, 6, 9, 10, 11, 12 – since the legend is the same for all three subplots you could include it only once. Furthermore please explain also each subplot in the figures captions.
Answer: Thanks for this good comment!
Each subplot has been added to the mentioned figures’ captions, and only one legend has been left for each mentioned figure.
- You can include the last section (Future works) in the section of Conclusions
Answer: Thanks for this good comment!
The Future work section has been included in the section of Conclusions.
Special thanks for your valuable time and wonderful comments.
Yours sincerely,
Best regards,
Nianxin Ren

Reviewer 2 Report
For the TLP system more details are needed (and not only references).
Give more details for the material used for this study (type, E, EA, Tbr). It is not clear that this system withstands stress (water depth 100m, t=0.04m).
Give the values of the restoring coefficients Kxx and Kzz of each tendon (or the total Kxx and Kzz).
For the Eq. (5), give more details for the εij.
Author Response
Dear editors and reviewers:
Thank you very much for the reviewer’s further comments concerning our manuscript. We have studied comments carefully and have made correction which we hope could meet with approval. Revised portion are marked in red in the paper. The main revision in the paper and the responses to the reviewer’s comments are listed as follows:
Responses to the comments of reviewer #2:
For the TLP system more details are needed (and not only references).
Answer: Thanks for this good comment!
The information of the steel tension legs’ E and σs has been added in Table 1.
Give more details for the material used for this study (type, E, EA, Tbr). It is not clear that this system withstands stress (water depth 100m, t=0.04m).
Answer: Thanks for this good comment!
The material of the tension leg is steel, with the E (elasticity modulus) =2.1×1011 N/m2 and σs (yield limit) = 3.45×108 N/m2. Based on the numerical results, the tension legs are safe for the modular system.
Give the values of the restoring coefficients Kxx and Kzz of each tendon (or the total Kxx and Kzz).
Answer: Thanks for this good comment!
For inner TLP modules, the restoring coefficients Kxx and Kzz (for one tension leg) can be estimated as follows:
;
where, Ft, E, A and l0 are the tension leg’s pretension force, the tension leg’s elasticity modulus, the tension leg’s section area and the tension leg’s length, respectively. The AQWA code can automatically calculate the corresponding Kxx and Kzz of the TLP. And more information for the design of the TLP can refer to reference [36].
For the Eq. (5), give more details for the εij.
Answer: Thanks for this good comment!
The Ɛij is the strain of the j-th tension leg of the i-th module.
Special thanks for your valuable time and wonderful comments.
Yours sincerely,
Best regards,
Nianxin Ren
